# Land Use Influences the Composition and Antimicrobial Effects of Propolis

**DOI:** 10.3390/insects13030239

**Published:** 2022-02-28

**Authors:** Amara J. Orth, Emma H. Curran, Eric J. Haas, Andrew C. Kraemer, Audrey M. Anderson, Nicholas J. Mason, Carol A. Fassbinder-Orth

**Affiliations:** 1Department of Biology, Creighton University, 2500 California Plaza, Omaha, NE 68178, USA; amaraorth@creighton.edu (A.J.O.); emmacurran@creighton.edu (E.H.C.); andrewkraemer@creighton.edu (A.C.K.); 2Department of Chemistry and Biochemistry, Creighton University, 2500 California Plaza, Omaha, NE 68178, USA; erichaas@creighton.edu (E.J.H.); nicholas.mason@cuanschutz.edu (N.J.M.); 3College of Engineering, University of Nebraska-Lincoln, 1400 R Street, Lincoln, NE 68588, USA; aanderson111@huskers.unl.edu

**Keywords:** propolis, honey bee, GC-MS, MIC_50_, land use, antimicrobial, honey bee pathogens

## Abstract

**Simple Summary:**

Honey bees collect a multitude of substances from plants, including nectar, pollen, and a lesser-known resin called propolis. Honey bees line their colonies with propolis to fill in cracks and potentially aid in their defense against pathogens such as fungi, bacteria, and viruses. Different plants contain different types of chemicals that are collected by bees to form propolis, and so one would expect the plants that bees visit to influence the quality of the propolis contained within honey bee colonies. This project explored the chemical composition and antibacterial effects of propolis collected from apiaries that were surrounded by different types of land use patterns in Iowa. Propolis samples collected from colonies that were surrounded by the highest levels of agriculture had the lowest abundance of chemical compounds and also the lowest antimicrobial activity detected for two of the bacteria species studied. These results add to a growing body of work that suggests that high intensity agricultural land use negatively impacts multiple aspects of honey bee colony health.

**Abstract:**

Honey bee propolis is a complex, resinous mixture created by bees using plant sources such as leaves, flowers, and bud exudates. This study characterized how cropland surrounding apiaries affects the chemical composition and antimicrobial effects of propolis. The chemical composition and compound abundance of the propolis samples were analyzed using Gas Chromatography-Mass Spectrometry (GC-MS) and the antimicrobial effects were analyzed using the 50% minimum inhibitory concentration (MIC_50_) assay against four relevant bee pathogens, *Serratia marcescens, Paenibacillus larvae, Lysinibacillus sphaericus,* and *Klebsiella pneumoniae*. Propolis composition varied significantly with apiary, and cropland coverage predicted mean sum abundance of compounds. The apiary with the highest cropland coverage exhibited significantly higher MIC_50_ values for *S. marcescens* and *K. pneumoniae* compared to other apiaries. These results demonstrate that agricultural land use surrounding honey bee apiaries decreases the chemical quality and antimicrobial effects of propolis, which may have implications for the impacts of land use on hive immunity to potential pathogens.

## 1. Introduction

While honey bees possess individual immunity, they also exhibit social immune responses to maintain health and reduce disease load at the colony level [1,2,3]. It has been proposed that cooperative immune behaviors evolved as a response to the limitations of prevailing innate or physiological bee immunity [2,4,5]. A common social immune behavior is the collection, deposition, and construction of a smooth 0.3–0.5 mm-thick “propolis envelope” coating the inside of beehive cavities by cementing bees [1,2]. Propolis is a resinous substance composed of sugars, wax, bee secretions, pollens, and various resins gathered by resin forager bees from local flora sources [1,2,6,7,8]. Although this natural resin executes an assortment of functions including maintaining a stable internal nest environment and preserving the physical integrity of the nest against cracks, uncontrolled airflow, and moisture, propolis is most highly regarded for its antipathogenic activity [1,2]. A strong propolis envelope provides seasonal benefits to colony immunity, and colonies with a thick propolis envelope exhibit higher rates of survival than those without propolis [9]. Chemical composition of propolis varies by bee species, region, season, and nearby floral diversity, and often contains flavonoids, terpenoids, and phenolics [7,8]. These highly prevalent compounds demonstrate antifungal, antibacterial, antiviral, and antioxidant activities [7,8,10,11,12,13,14,15].

Honey bee propolis has been utilized for its antimicrobial properties in holistic human medicine since 300 BCE [14,16] and the antimicrobial effects of propolis have been analyzed in over 625 bacteria and fungi that inhabit the human microbiome [10,17,18,19,20,21,22,23]. However, there are more limited reports of the role of propolis on bee pathogens [1,2,8]. The goal of this study was to investigate the putative impact of land use on propolis antibiotic activity against *Serratia marcescens, Klebsiella pneumoniae, Paenibacillus larvae*, and *Lysinibacillus sphaericus*, selected for their potential impact on honey bee populations.

*S. marcescens* is an opportunistic gram-negative pathogen in many animals, including insects [24]. It is commonly found in the guts of honey bee workers, and some strains are pathogenic, resulting in bacterial escape from the midgut, proliferation in the hemolymph, and fatal septicemia [15,25].

*K. pneumoniae* is a gram-negative bacterium commonly found in honey bees. It is a dominant species in the aerobic microflora in the Asian honey bee (*Apis cerana indica* [26]) and may play a role in early stages of development of *P. larvae* infection in the European honey bee (*Apis mellifera* [27]). *P. larvae* is a notorious gram-positive honey bee larvae pathogen and the causative agent of American foulbrood (AFB) [25,28,29]. This pathogen is transmitted by nurse bees feeding *P. larvae* spore-contaminated food to naive larvae, where the pathogen is able to germinate and proliferate in the midgut lumen. Eventually, the *P. larvae* bacteria breach the midgut epithelium and transition into an invasive stage in the hemocoel, killing and decomposing the larvae host, and sporulating for future transmission [28,29]. Despite human efforts to control *P. larvae* by burning infected hive material and diseased colonies, *P. larvae* remains an exceptionally destructive honey bee pathogen [25,29].

*L. sphaericus* is a gram-positive, rod-shaped bacterium found in soil, aquatic habitats, plant material, and insect cadavers [30]. It produces a suite of lethal virulence factors known to be insecticidal, and has recently been recognized as the etiological agent of an emerging brood disease in the Australian crop pollinating, stingless bee, *Tetragonula carbonaria* [25,30].

We used Gas Chromatography-Mass Spectrometry (GC-MS) to investigate the chemical composition and diversity of propolis samples collected from hives from five Iowa apiaries with different land-use geography. We then used the 50% minimum inhibitory concentration (MIC_50_) assay to analyze the antimicrobial effects of propolis samples on four selected honey bee bacteria pathogens. We predicted that higher levels of cropland surrounding apiaries would affect the chemical composition of propolis and decrease its antimicrobial activity.

## 2. Materials and Methods

### 2.1. Propolis Collection and Extraction

Propolis was collected in October 2019, from five different apiaries of a commercial beekeeping operation in Fayette and Clayton counties in Iowa. Samples were collected from 5–10 colonies at each location, yielding a total of 45 samples. Propolis was collected from the front rim of the top lid of each hive. This location on the hive was chosen because it is the area of most active recent propolis placement by honey bees, due to frequent beekeeper disturbance. Each propolis sample was collected into sterile plastic tubes and stored at −80 °C until further use. Propolis samples were then pulverized with a mortar and pestle, and diluted in 100% ethanol (0.2 g propolis in 1.8 g ethanol). Samples were then mixed at 500 rpm at 30 °C for 48 h in the dark. Samples were filtered using a 0.22 μm syringe filter (Fisher Scientific, Hampton, NH, USA). Filtered samples were then diluted according to specific needs for chemical or antimicrobial analysis.

### 2.2. Gas Chromatography Mass Spectroscopy (GC-MS)

Propolis samples were analyzed by Gas Chromatography Mass Spectroscopy using an Agilent 6850 Series Gas Chromatography—Agilent 5975 Series Mass Spectrometer (Agilent Technologies, Santa Clara, CA, USA). Briefly, propolis samples were added to scintillation vials, placed in the Agilent GC-MS machine and 1 μL was injected onto a 30 m × 250 μm × 0.25 μm 5% Phenyl Methyl Siloxane column at an initial temperature of 60 °C increased 3.0 °C/min to 240 °C. The temperature was held for 20 min. The results were analyzed using Agilent’s ChemStation version E.02.02.1431 and NIST Mass Spectral Search Program Version 2.0 [31].

The chemical identity of each major compound in the propolis samples was determined by comparing the apex of individual component spectrums to the NIST Mass Spectral Library. Each of the major components identified by the NIST library also shared similar retention times across all the propolis samples, further confirming the identities of the compounds. The area of each peak was reported as a percent of the total ion chromatogram. The area percent of each identified compound in one propolis sample was compared via percent ratios to the same compound of the other propolis samples. Only those compounds with consistent NIST library recognition across all spectra were recorded. It is important to note that some constituents remained unidentified due to the lack of library spectra for the corresponding compounds. Future investigation into novel compounds found in propolis is warranted.

### 2.3. Land Use Analysis

The Cropland Data Layer (CDL) product of the U.S. Department of Agriculture (USDA) National Agricultural Statistics Service was used to assess land cover surrounding the apiaries. The CDL is a crop-specific land cover map with a group resolution of 56 m [32]. Land use was assessed in an area encompassing a 3.2 km radius from each apiary [32]. This size was chosen based on the previous research indicating that foragers generally stay within a 3.2 km (2 mile) radius from their hive [33]. For this study, cropland use was quantified out of the total land use. Cropland included: alfalfa, corn, millet, oats, rye, sorghum, soybean, and winter wheat.

### 2.4. MIC_50_

The antimicrobial effects of the propolis samples were measured by performing a MIC_50_ assay, according to Wiegand et al. [34] with modifications. Briefly, propolis samples were serially diluted with phosphate buffered saline (PBS) to yield a diluted range of 0.1 mg propolis/mL to 17 mg propolis/mL, a range previously observed to be inhibitory to bacterial growth [7,35,36]. Ethanol serially diluted in PBS was used as the negative control. Fifty μL of each propolis sample or negative control dilution was then plated on a 96 well plate in triplicates.

*Paenibacillus larvae* (ATCC 9545) and *Lysinibacillus sphaericus* (ATCC 4525) were obtained from American Type Culture Collection (ATCC, Manassas, VA, USA), and *Serratia marcescens* and *Klebsiella pneumoniae* were obtained from Carolina Biological Supply Company (Burlington, NC, USA). Bacterial suspensions were prepared by transferring two colonies of bacteria from agar media into tryptic soy broth (*S. marcescens* and *K. pneumoniae*) or brain heart infusion broth (*P. larvae* and *L. sphaericus*). Viability of bacteria was determined on a hemocytometer using erythrosine B. A 1 × 10^5^ cells/ml bacterial solution was then prepared using the appropriate liquid broth medium. Next, 100 μL of each bacteria suspension was added per well in a 96 well plate. The plates were then incubated at 30 °C (*S. marcescens* and *L. sphaericus*) and 37 °C (*K. pneumoniae* and *P. larvae*) for 24 h. Following incubation, 0.75 μL from each well was diluted in 50 μL of PBS and transferred to a well in a 12 well plate with the appropriate agar media for each bacterium. These plates were then incubated at 30 °C (*S. marcescens* and *L. sphaericus*) and 37 °C (*K. pneumoniae* and *P. larvae*) for 24 h. Colonies of bacteria were then counted and the MIC_50_ of each sample for each bacterium was determined as the lowest propolis concentration to inhibit 50% of the bacterial growth.

### 2.5. Statistical Analyses

#### 2.5.1. Relationship between Land Use and Propolis Production

In this analysis, we performed a linear regression of land use and propolis production, using the proportion of the landscape dedicated to seasonal crops surrounding each apiary as the explanatory variable and the mean peak sum abundance of propolis produced among hives at that apiary as the response variable.

#### 2.5.2. Comparison of Propolis Compounds among Apiaries

We performed a principal component analysis (PCA) on the results of the GC-MS dataset to obtain a set of independent variables that together described variation in the relative composition of compounds found in each hive’s propolis. We then used the ‘lm.rrpp’ function from the ‘RRPP’ package in R [37,38] to perform a permutational ANOVA, using apiary as the explanatory variable and the variables produced with the PCA serving as the response data. Since it was possible that our distance measure used for the PCA (Euclidean) affected our results, we conducted the same analysis using a Jaccard distance measure. We present the Euclidean analysis here since our results were concordant across distance measures. See Appendix A for pairwise comparison values.

#### 2.5.3. Effectiveness of Propolis against Pathogens

In this analysis, we constructed a series of linear models between apiary and MIC_50_ scores, using apiary as the explanatory variable and the MIC_50_ score for each hive’s propolis against each pathogen as the response variable (resulting in one linear model for each pathogen). In the event of significant models, we used a Tukey HSD test to identify which apiaries were significantly different from one another. R script for all analyses performed can be found in Appendix A.

## 3. Results

Propolis samples from five apiaries in northeast Iowa were analyzed for their chemical composition and antimicrobial effects against four relevant bee pathogens. Land use surrounding these apiaries was also assessed to determine potential relationships between propolis quality and land use.

### 3.1. Relationship between Land Use and Propolis Production

Cropland coverage predicted the sum abundance of propolis (F_1,3_: 29.23; adjusted R^2^: 0.876; *p* = 0.012; Figure 1). Apiary A had the highest percent cropland coverage (91%), while Apiary E had the lowest cropland coverage (22%).

### 3.2. Comparison of Propolis Compounds among Apiaries

Apiary was a significant predictor of the relative distribution of propolis compounds (F_4,19_ = 6.25; R^2^ = 0.57; *p* < 0.001; Figure 2). The first and second principal components (PC1 and PC2) of total variation are visualized in Figure 2, representing 70.8% and 13.7% of the total variation, respectively. Points close to one another had similar chemical profiles, while those far apart differed with respect to the identity and relative proportions of chemicals identified. Results of a pairwise analysis showed that the propolis compounds of Apiary C differ from all other apiaries, Apiaries A and D were significantly different, while all other comparisons were non-significant (Figure 2, inset; Appendix A).

The most abundant compounds in each apiary are shown in Table 1 and Appendix A. Benzoic acid was the most prevalent compound found in all apiaries, followed by coumaran and 2-methoxy-4-vinylphenol.

### 3.3. MIC_50_

#### 3.3.1. *Klebsiella pneumoniae*

Apiary was a significant predictor of MIC_50_ against *K. pneumoniae* (F_4,32_ = 4.33; adjusted R^2^ = 0.27; *p* = 0.007). In particular, the MIC_50_ for Apiary A was significantly larger than most other apiaries (Tukey HSD A vs. B: difference of the means = 1.54, *p* = 0.010; A vs. C: difference of the means = 1.63, *p* = 0.006; A vs. D: difference of the means = 1.69, *p* = 0.021). This indicates that the propolis for Apiary A had the lowest antimicrobial activity against *K. pneumoniae* out of the five apiaries tested (Figure 3).

#### 3.3.2. *Serratia marcescens*

Apiary was a significant predictor of MIC_50_ against *S. marcescens* (F_4,32_ = 4.33; adjusted R^2^ = 0.18; *p* = 0.032). In this case, Apiary A was significantly larger than only Apiary C (Tukey HSD: difference of the means = 1.72, *p* = 0.020, Figure 3).

#### 3.3.3. *Lysinibacillus sphaericus* and *Paenibaccillus larvae*

Apiary was not a significant predictor of MIC_50_ against *L. sphaericus* (F_4,32_ = 1.17; adjusted R^2^ = 0.02; *p* = 0.345) nor *P. larvae* (F_4,32_ = 0.28; adjusted R^2^ = −0.03; *p* = 0.890, Figure 3). Although no significant differences were found among apiaries, it is important to note that propolis was more effective against killing *P. larvae* compared to all other bacteria (*p* < 0.003, Figure 4).

## 4. Discussion

The presence of cropland surrounding apiaries negatively affected propolis composition and antimicrobial activity in this study. In general, propolis samples exhibited high levels of antimicrobial activity against all four bacteria studied: *S. marcescens*, *K. pneumoniae*, *P. larvae*, and *L. sphaericus*. Propolis was notably most effective against *P. larvae* followed by *L. sphaericus*. This finding corroborates previous research indicating higher antimicrobial activity against gram positive compared to gram negative bacteria [10,13,39]. Multiple studies have also shown propolis is highly effective at inhibiting the growth of *P. larvae* at similar levels found in this study [40,41].

To our knowledge, this is the first report of the effectiveness of propolis on *L. sphaericus*, an important pathogen of stingless bees [30]. Stingless bees are known to produce propolis [42], and high quality propolis could therefore be important in the stingless bee’s social defenses against *L. sphaericus*.

The most prevalent component of propolis gathered in this study was benzoic acid, a weak aromatic acid that likely contributes to the antimicrobial properties of propolis through acidification of bacterial cytoplasm [43,44]. Other significant propolis components included coumaran (dihydrobenzofuran), 2-methoxy-4-vinylphenol, and trans-cinnamic acid. Trans-cinnamic acid is also a weak acid with antimicrobial activity similar to benzoic acid [44]. Coumaran and 2-methoxy-4-vinyl-phenol exhibit antimicrobial activity by inhibiting bacterial DNA replication and bacterial wall lipoproteins dynamics [45,46].

This study provides novel insight into the potential relationship between the composition and antimicrobial activity of propolis and land use. Propolis composition has been found to be unique to the geographical area and source plant diversity [13,47,48,49,50,51]. Propolis composition likely directs antimicrobial characteristics: a greater diversity of resin sources present in propolis provides functional redundancy in its antimicrobial activity [41,51,52]. Differences in geography have been demonstrated to produce varied propolis chemical profiles and biological activity. For example, propolis collected from various regions of Europe have significantly different chemical profiles [6,7]. Furthermore, there appears to be a positive correlation between antioxidant activity and total phenolic and flavonoid content according to different geographical origins in Europe [7]. Wilson et al. also determined that propolis collected from different regions of the United States had different chemical compositions and different patterns of antimicrobial activity against *P. larvae* and *Ascosphaera apis* [41]. While some studies have explored the specific sources of propolis constituents, such as poplar trees [53], the species-specific plant origins for most propolis constituents are unknown. Further investigation of specific plant species contributions to propolis is needed to potentially use propolis as a bioindicator for ecosystem stability and plant diversity of a region. It is also noteworthy that the significant differences in propolis composition detected among apiaries in this study were found within a 300 square mile area across two counties in northeast Iowa, indicating that propolis composition and activity can vary across a much smaller geographic range than has been previously studied.

## 5. Conclusions

Intensive agriculture is increasingly being investigated for its negative impacts on bee health [54]. While row crops can provide short term nutrition, they cannot sustain colonies through different seasons, and the lack of biodiversity in intensively farmed areas likely contributes to poor colony nutrition and colony decline [55,56,57]. Row crops like corn and beans are also known sources of pesticide exposure, such as neonicotinoid insecticides and fungicides. Exposure to pesticides in corn fields reduces colony health [58], and neonicotinoids have been detected in honey bee propolis [59]. A transition from grassland to row crops like soybeans is also detrimental to honey yield [60]. While there are many chemical compounds that are common to all propolis samples collected (e.g., benzoic acid), greater plant diversity is expected to produce more diverse compounds in the propolis generated from these plants, a result that was confirmed in this study. Therefore, a monoculture of cultivated crops not only produces lower nutritional sources for the bees, but also produces less propolis compound diversity and lower propolis antimicrobial activity as well. In this study, we determined that the presence of row crops near honey bee apiaries decreased the quality of honey bee propolis, adding to the growing body of evidence that intensive row crop agriculture can have negative impacts on honey bee health.

## Figures and Tables

**Figure 1 insects-13-00239-f001:**
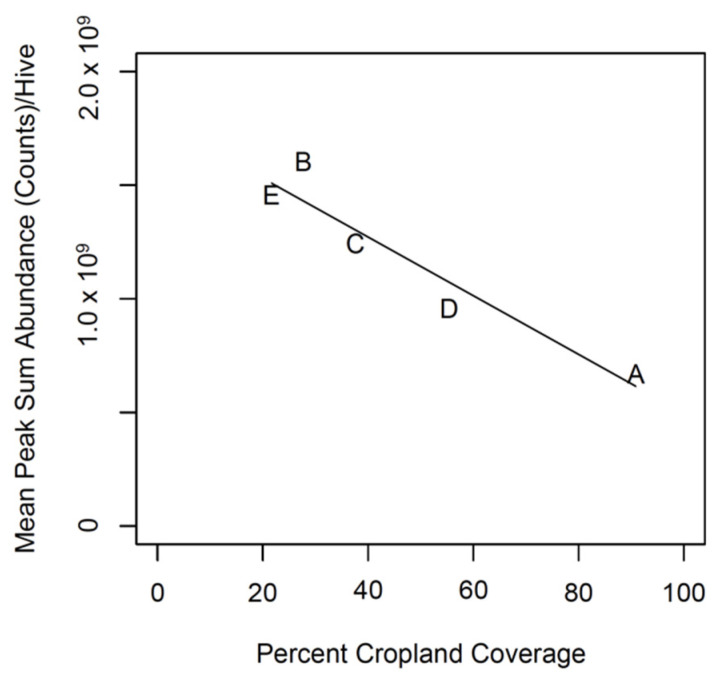
Mean peak sum abundance vs cropland coverage. Mean peak sum abundance was obtained from GC-MS and represents the summed counts of different compounds in the propolis. Cropland coverage for a 3.2-km radius around each apiary (A–E) was obtained from the Cropland Data Layer product of the USDA NASS.

**Figure 2 insects-13-00239-f002:**
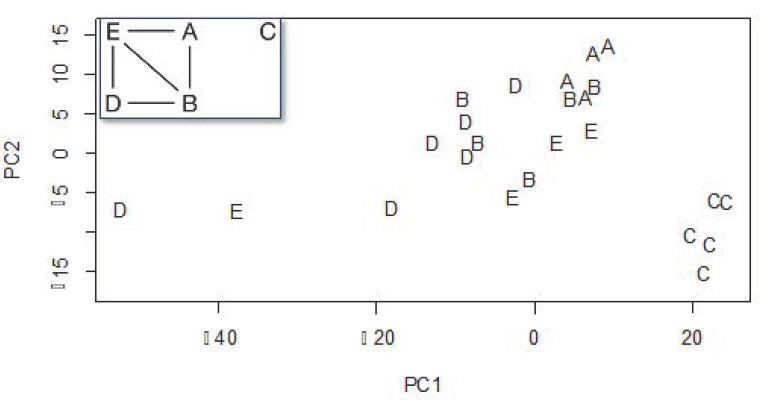
Principal components plot of compound variation among propolis samples (main figure), and pairwise analysis network graph (inset). Apiaries are represented by letter (A–E).

**Figure 3 insects-13-00239-f003:**
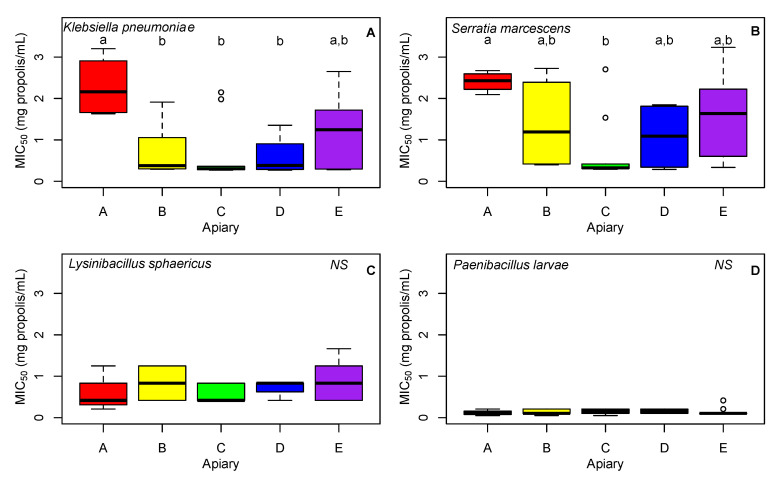
Box plot of MIC_50_ values of propolis samples for four honey bee pathogens, *K. pneumoniae* (**A**), *S. marcescens* (**B**), *L. sphaericus* (**C**), and *P. larvae* (**D**). Apiaries are represented by letter (A–E). Boxes not sharing the same letter superscript are significantly different from each other (*p* < 0.05). Note that no apiaries differed for *L. sphaericus* or *P. larvae* (NS).

**Figure 4 insects-13-00239-f004:**
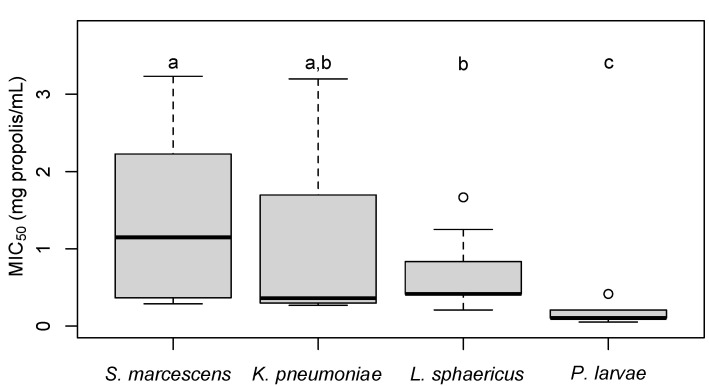
Box plot of MIC_50_ values of propolis samples according to pathogen tested. Boxes not sharing the same letter superscript are significantly different from each other (*p* < 0.05). This section may be divided by subheadings. It should provide a concise and precise description of the experimental results, their interpretation, as well as the experimental conclusions that can be drawn.

**Table 1 insects-13-00239-t001:** Top ten most abundant compounds in propolis samples, according to apiary.

Rank	Apiary A	Apiary B	Apiary C	Apiary D	Apiary E
1	benzoic acid (63%)	benzoic acid (53%)	benzoic acid (60%)	benzoic acid (40%)	benzoic acid (42%)
2	coumaran (15%)	coumaran (17%)	coumaran (20%)	coumaran (29%)	coumaran (18%)
3	2-methoxy-4-vinylphenol (8%)	2-methoxy-4-vinylphenol (7%)	2-methoxy-4-vinylphenol (7%)	2-methoxy-4-vinylphenol (11%)	trans-cinnamic acid (10%)
4	3-methoxyaceto-phenone (7%)	alpha bisabolol (5%)	benzyl alcohol (4%)	3-methoxyaceto-phenone (9%)	2-methoxy-4-vinylphenol (7%)
5	benzyl alcohol (6%)	benzyl benzoate (5%)	benzyl benzoate (2%)	benzyl alcohol (7%)	alpha bisabolol (5%)
6	benzyl benzoate (2%)	benzyl alcohol (3%)	alpha bisabolol (1%)	alpha bisabolol (3%)	curcumene (4%)
7	cedrane (1%)	trans-cinnamic acid (3%)	acetophenone (1%)	benzyl benzoate (2%)	benzyl benzoate (3%)
8	2-benzoylaminoethanol (1%)	naphthalene (2%)	curcumene (1%)	trans-isoeugenol (2%)	vanillin (3%)
9	4-benzyloxybenzoic acid (1%)	vanillin (1%)	trans-isoeugenol (1%)	trans-cinnamic acid (1%)	benzyl alcohol (3%)
10	4-hydroxy-3-methylacetophenone (1%)	acetophenone (0.4%)	vanillin (1%)	acetophenone (1%)	alpha farnesene (1%)

## Data Availability

Data are available through Dryad at: doi:10.5061/dryad.t4b8gtj2n.

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
