# Peer review of "Land Use Influences the Composition and Antimicrobial Effects of Propolis"

_insects, 2022, doi:10.3390/insects13030239_

Round 1
Reviewer 1 Report
Dear Authors,
congratulation to excellent scientific work.
The aim of the paper is interesting and well selected. Propolis is important bee product and one of the most strong medicine from them. However, there is lack of scientific papers and studies concernig the chemical composition of propolis as well as influence of surrounding on its compositon. All parts of the paper - Introduction, Materials and Methods, Results, Discussion, References - are appropriate. The style of the language is fresh and good-to-understand. I have only a few comments/suggestions (mainly formal):
- line 60: .... since 300 BCE (14, 16). And....
- line 96: ... in October .... (add the year)
- line 112: ....of 60 °C ... (space between the number and unit, edit it in whole text, also line 151)
- line 133: ...according to Wiegand et al. (2008)...
- table 1: edit in - the same font type like text, ...
- figure 2: I suggest the simple title and explanations as descriptions under the figure. Check it in the Instructions for authors and if you decide to change it, edit it in whole paper.
- line 210-226: write the latin names of bacteria always (including names of subtitles) by italic and MIC50 with 50 as upper index
- figure 4: correct the x (axis) to legible bacterial names
- line 266: Wilson et al. also... (dot)
- line 283: ...plants--a result... (edit--)
- general note: Consider the adding of description of potential sources of propolis in evaluated area in some place of the paper (Introduction, Methodology or Results/Discussion).
Kind regards and good luck in next research,
VK
Reviewer 2 Report
The study by Orth et al. provides novel insight into the probable relationship between the composition and antimicrobial effect of honeybee propolis and land use, and demonstrated that, a monoculture of cultivated crops not only produces lower nutritional sources for the bees, but also produces less propolis compound diversity and reduced propolis antimicrobial effect. This was evidently showed in this study, where the presence of row crops near honeybee apiaries decreased the quality of propolis, and lent credence to the fact that, intensive row crop agriculture may negatively impact the overall well-being of honeybee.
Nevertheless, it would be appreciated if the GCMS chromatograms for each propolis sample analyzed are included either in the manuscript or as supplementary file to appreciate the detailed constituents and features of the analytes identified. Also, what was the degree of accuracy and superimposable factor of the NIST library used in the identification of the constituents of the propolis?
